# Expand, Highlight, Generate:
# RL-driven Document Generation for Passage Reranking

**Arian Askari[1], Mohammad Aliannejadi[2], Chuan Meng[2]**
**Evangelos Kanoulas[2], Suzan Verberne[1]**
[1]Leiden University {a.askari,s.verberne}@liacs.leidenuniv.nl
[2]University of Amsterdam {m.aliannejadi,c.meng,e.kanoulas}@uva.nl

## Abstract

Generating synthetic training data based on large language models (LLMs) for ranking models has gained attention recently. Prior studies use LLMs to build pseudo query-document pairs by generating synthetic queries from documents in a corpus. In this paper, we propose a new perspective of data augmentation: generating synthetic documents from queries. To achieve this, we propose DocGen, that consists of a three-step pipeline that utilizes the few-shot capabilities of LLMs. DocGen pipeline performs synthetic document generation by (i) expanding, (ii) highlighting the original query, and then (iii) generating a synthetic document that is likely to be relevant to the query. To further improve the relevance between generated synthetic documents and their corresponding queries, we propose DocGen-RL, which regards the estimated relevance of the document as a reward and leverages reinforcement learning (RL) to optimize DocGen pipeline. Extensive experiments demonstrate that DocGen and DocGen-RL significantly outperform existing state-of-the-art data augmentation methods, such as InPars, indicating that our new perspective of generating documents leverages the capacity of LLMs in generating synthetic data more effectively. We release the code, generated data, and model checkpoints to foster research in this area[1].

## 1 Introduction

Data augmentation for information retrieval (IR) has gained attention as a promising research area. Previous studies use large language models (LLMs) (Zhao et al., 2023) to generate synthetic training data for retrievers (Jeronymo et al., 2023; Dai et al., 2023; Boytsov et al., 2023; Bonifacio et al., 2022), significantly improving effectiveness of unsupervised retrievers. Specifically, these studies all build pseudo query–document pairs by generating synthetic queries given documents in an existing corpus.

However, previous work suffers from the following limitations: (i) generated synthetic queries tend to have a larger lexical overlap with relevant documents than human queries, leading to a mismatch between synthetic and real data; (ii) generating queries from a document in an existing corpus cannot cover queries for which the relevant information is spread over multiple documents; moreover, in reality, there could be difficult and complex queries for which there are very few to zero relevant documents in existing corpora; thus a retriever trained on such data is less likely to learn to retrieve effectively for complex queries.

To overcome these limitations, we propose to generate synthetic documents from given queries,[2] This paradigm has three benefits: (i) taking user queries as input makes the resulting training data more representative of real information needs; (ii) generating synthetic documents enables us to create relevant documents for complex queries that have few or no relevant documents in an existing corpus; LLMs are pre-trained on a vast number of corpora, and have the potential in generating relevant documents for complex queries by aggregating relevant information they have seen during pre-training; (iii) Yu et al. (2023) show that generating a document is closer to the language modeling objective of the LLM pre-training than generating queries, thus generating documents can better utilize the knowledge stored in LLMs parameters, producing higher-quality text.

We use the few-shot capabilities of LLMs (Dong et al., 2022) to generate synthetic documents given queries. Our preliminary experiments identified two challenges: (i) feeding raw queries to an LLM could result in low-quality synthetic documents[3];

---

[1]https://github.com/arian-askari/docgen

[2]In this paper, we interchangeably use the words 'document' and 'passage.'

[3]This observation is based on open-source LLMs, which

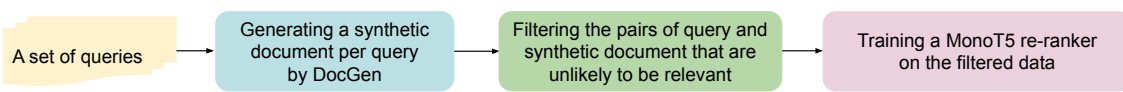

Figure 1: The illustration of training passage reranking using the DocGen.

we think it is because raw queries contain limited helpful information to trigger the knowledge stored in an LLM; and (ii) it is important to ensure the close relevance between a query and its corresponding synthetic document. To address the first challenge, we propose a three-stage pipeline for document generation, named DocGen pipeline, that effectively uses the few-shot capabilities of LLMs. The idea of the *DocGen pipeline* is to enrich raw queries before generating synthetic documents to effectively trigger the knowledge stored in an LLM. Specifically, the DocGen pipeline consists of three steps: (i) *query expansion*, which expands a raw query to clarify the information need, (ii) *query highlighting*, which highlights the important terms of the expanded query, and (iii) *synthetic document generation*, which generates the final synthetic document given the query after expansion and highlighting. To solve the second challenge, we propose DocGen-RL, which regards the relevance between a generated document and its corresponding query as a reward function, and propose to leverage reinforcement learning (RL) to optimize the *DocGen pipeline*. Our preliminary experiments show that *query highlighting* is a challenging step; ineffective highlighting, i.e., highlighting words that are not semantically the most important words in the query, has a negative impact on the final relevance. Therefore, DocGen-RL optimizes *query highlighting* stage through RL.

The commonly used ranking pipeline consists of a first-stage retriever, e.g., BM25 (Robertson and Walker, 1994), that efficiently retrieves a set of documents from the full document collection, followed by one or more rerankers (Nogueira et al., 2020) that improve the initial ranking. Currently, the most effective rerankers are T5-based rankers with a sequence-to-sequence architecture. In this paper, we refer to these rerankers as MonoT5. In the common reranking set-up, BM25 (Robertson and

Walker, 1994) is widely leveraged (Anand et al., 2021; Nogueira et al., 2020) for finding the top-$k$ documents to be reranked. In this work, we focus on data augmentation for the reranking stage where BM25 is used for first-stage passage retrieval and MonoT5 is used for reranking passages.

Our main contributions are as follows:

- We propose a new direction on data augmentation for IR: generating synthetic documents from queries. We devise the *DocGen pipeline*, a three-step few-shot learning-based pipeline, which improves the quality of synthetic document generation by expanding and highlighting queries.
- To enhance the relevance between generated synthetic documents and their corresponding queries, we propose DocGen-RL, which regards the relevance as a reward and leverages RL to optimize the *DocGen pipeline*.
- We conduct a comprehensive analysis of the proposed methodology. Through rigorous evaluation and experimentation, we thoroughly examine the impact of each step of the *DocGen pipeline* and RL training, shedding light on their strengths, limitations, and potential areas of improvement.

## 2 Related Work

**Data augmentation for IR using LLMs.** In-Pars (Bonifacio et al., 2022), Promptagator (Dai et al., 2023), and InPars-v2 (Jeronymo et al., 2023) utilize LLMs to generate synthetic queries for given documents. Particularly, InPars-v2 achieves state-of-the-art results on the BEIR benchmark (Thakur et al., 2021) by using an open-source language model, GPT-J-6B (EleutherAI, 2023), for query generation and a powerful external reranker, MonoT5-MS MARCO (Nogueira et al., 2020), to filter the top-10k high-quality pairs of synthetic query–document pairs. Pseudo queries have also been used beyond data augmentation. For instance, Abolghasemi et al. (2023) leverages the pseudo queries for the estimation of retrievability bias in document retrieval.

Synthetic document generation given queries for data augmentation has not been explored in prior work. The recent work by Askari et al. (2023a,b) compares the effectiveness of models trained by

---

contrasts with closed-source LLMs used for commercial purposes, such as ChatGPT and GPT 3.5/4.0. These closed-source LLMs seem capable of generating higher-quality text directly from raw queries. However, we do not consider them to ensure our research remains independent from commercial organizations. Also, for closed models, it is unclear if there is data leakage between their pre-training data and the datasets we used for evaluation in this paper.

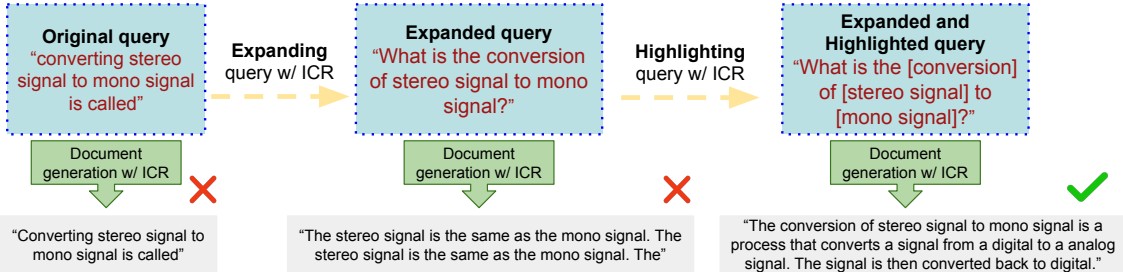

Figure 2: The DocGen's pipeline: query expansion, query highlighting, and synthetic document generation. Few-shot examples (three) are provided at each step.

ChatGPT-generated and human-written passages. That study has two limitations: ChatGPT is a closed-source LLM, and they only investigate the zero-shot setting of the LLM. In contrast, we fully investigate the effectiveness of open-source LLMs with using few-shot (three) examples in a unified framework for modeling the task of data augmentation for IR models. Gao et al. (2022) generates a hypothetical document and encodes the document into an embedding vector by an unsupervised contrastively learned encoder to retrieve similar real documents based on vector similarity.

Generating documents from queries is also explored by Yu et al. (2023). They propose GenRead for the open-domain question answering task, achieving state-of-the-art performance. GenRead first prompts an LLM to generate contextual documents based on a given question, and then reads the generated documents to produce the final answer. Our work differs with GenRead as we focus on data augmentation for passage reranking, however, we can use GenRead as a strong baseline for generating documents for given queries to build synthetic query–document pairs. Recently, (Gabburo et al., 2023) propose data augmentation techniques to improve a generative QA model using supervision from automatic question answering evaluators.

**Reinforcement learning on LLMs.** RL has been applied to enhance models in various natural language processing tasks. These tasks include machine translation (Wu et al., 2016; Kiegeland and Kreutzer, 2021), summarization (Stiennon et al., 2020; Paulus et al., 2017), dialogue systems (Jaques et al., 2020), question generation (Pang and He, 2021), and more. Ramamurthy et al. (2023) gives an extensive overview of research on RL for LLMs up to 2022. Li et al. (2023) train a policy language model with RL leading to significant performance improvements across summarization and dialogue response generation with

limited training data. Ramamurthy et al. (2023) present RL4LMs library for faciliating training generative language models with RL and show how controlling the stability of RL could achieve high effectiveness for training LLMs. They implement proximal policy optimization (PPO)[4] as the policy gradient method as recent works have shown PPO to be strictly superior to reinforce in multiple domains. To the best of our knowledge, there is no work on using RL on LLMs for IR data generation.

## 3 Synthetic Document Generation

In this section, we present the details of our proposed method, DocGen, and its reinforcement learning extension, DocGen-RL.

**Task definition.** Given a query $q \in Q$, where $Q$ is a set of queries, the goal of synthetic document generation is to generate a document $d$, such that $d$ is relevant to $q$, and use $C = \{d, q\}$ collection to train a retriever or reranking model.

### 3.1 DocGen

We tackle the document generation task for IR, specifically the passage reranking task. DocGen consists of three main components illustrated in Figure 1: (i) **document generation:** *DocGen pipeline* that generates an initial set of synthetic documents for given queries; (ii) **consistency filtering:** it filters out the low-quality query–document pairs, using a consistency filtering approach (Dai et al., 2023); and (iii) **reranking:** it fine-tunes a MonoT5 reranker on the filtered synthetic data. Next, we detail different components of the *DocGen pipeline*.

### 3.1.1 Few-shot Guided Document Generation

As illustrated in Figure 2, the document generation pipeline of DocGen itself consists of three steps,

---

[4]We refer readers to (Schulman et al., 2017) for a detailed explanation.

namely, (i) query expansion; (ii) query highlighting; and (iii) document generation. We implement each of the three steps using a prompt composed of three examples for few-shot learning.

**Prompt design.** Each prompt is the concatenation of a prefix $P_i$ and a query $q$ where the prefix $P_i$ is the prompt template for step $i$ and consists of $N$ pairs of examples, e.g., for query expansion, the $P_i$ equals $\{(q_1, q_1^*), ..., (q_N, q_N^*)\}$ where $q_N$ refers to the original query and $q_N^*$ refers to the expanded query. For each step, we manually design a prompt, which is always the same regardless of the input query $q$, i.e., we can potentially generate millions of synthetic training examples using only $N$ manually annotated examples. Given a query $q$ in step $i$, we feed $P_i||q$ to the LLM that generates the desired output.

**Query expansion.** Our preliminary experiments reveal that query expansion and highlighting improve the quality of the generated documents. We observe that expanding short keyword-based queries to longer queries in the form of natural language while highlighting the important words enhances the ability of LLMs to generate documents with higher quality. It is noteworthy that using state-of-the-art IR methods on query expansion is out of the scope of our study since we do not apply query expansion to improve document retrieval. Instead, our goal is to re-construct the query in natural language for which the LLM can generate documents more effectively. The prompt template can be seen in Figure 3 in the appendix.

**Query highlighting.** Weller et al. (2023) shows that using specific terms or phrases can trigger the LLM's memory to manipulate its output. For example, in their work they show that adding the phrase "As Wikipedia indicates" in the prompt manipulates the LLM to recall the text from Wikipedia more often. Similarly, we find that there are terms in BLOOM's training data (Scao et al., 2022) that are highlighted with square brackets. Therefore, inspired by Weller et al. (2023) finding and our preliminary analysis, we highlight important words of the query using square brackets to manipulate the LLM to pay more attention to those words while it considers the expanded query in the document generation process. As demonstrated in Figure 2, we highlight the important words of a query by adding square brackets around them, e.g., "What is the [conversion] of [stereo signal] to [mono signal]?" Examples of this prompt template and analysis on

optimal character for highlighting can be seen in Figure 4 and Section A.3 in the appendix.

**Document generation.** For few-shot document generation, we use the same examples – with modified queries that are expanded and highlighted – from the prompt template provided by Bonifacio et al. (2022). The InPars prompts consist of three examples, each containing a query and a relevant document. The prompt template can be seen in Figure 5 in the appendix.

### 3.1.2 Consistency Filtering

To ensure the quality of the generated documents we apply the common consistency filtering approach, originally proposed for synthetic query generation (Dai et al., 2023). Consistency filtering has been proved crucial for synthetic data generation on QA tasks (Lewis et al., 2021). Dai et al. (2023) shows that consistency filtering based on the generated data alone can work well. Inspired by these works, we first use the expanded and highlighted query and synthetically generated document pairs to train an initial retriever which we call MonoT5-CF. Given a query–document pair $(q, d)$, we use the MonoT5-CF to predict the most relevant passages for $q$. We keep the query–document pair in the final dataset, only when $d$ is the top-1 document returned by the retriever. One can argue that this technique is flawed because the filtering model (MonoT5-CF) is trained on the same noisy synthetic data that it intends to filter. However, Dai et al. (2023) show this filtering technique substantially reduces the number of synthetic data and significantly improves retrieval performance.

### 3.1.3 Reranking

We fine-tune a MonoT5 on the filtered data from the previous step and call it *DocGen reranker*.

### 3.2 RL-guided Document Generation

DocGen-RL is motivated by the challenges we encounter during query highlighting in few-shot guided document generation. We find out that when using few-shot examples, the model occasionally unreasonably highlights words that are not semantically the most important words of the query. For example, in some cases stop words, question marks, or other punctuation marks are highlighted. It is important to emphasize that highlighting the query words is not a straightforward task with a single correct solution. The ultimate goal is to highlight query words that lead to generating higher-quality

documents. Given the nature of the task and its dependence on the document generation module, one naive solution would be to try all possible highlighting combinations and find the optimum, which is an NP-hard problem.

To address this challenge, we employ RL to ensure and improve the robustness of query highlighting and ultimately enhance the quality of document generation. During RL training, we optimize the process of query highlighting with the goal of generating documents of higher quality that are more relevant to the highlighted query. To achieve this, we pass the highlighted query to the LLM, generate a document with few-shot learning, and use the predicted relevance by DocGen as the reward for the LLM that highlights the query. We optimize the LLM for highlighting while retaining the few-shot examples of the prompt during RL training. By using the *DocGen reranker* as our reward function, we make our reward function completely based on few-shot learning with three examples, dependent only few shots (three examples), as we train *DocGen reranker* without any supervision of the actual dataset, i.e., MS MARCO.

### 3.2.1 Optimization Objective

Our goal is to optimize the query highlighting tasks in an RL setup by a policy language model (PLM) (Schulman et al., 2017) in order to improve the quality of LLM's synthetic document generation. To achieve this, we maximize the relevance measure, $\mathcal{R}$, between the query after expansion and highlighting, and the generated document, $\boldsymbol{y}$. The relevance measure $\mathcal{R}$ is the predicted relevance score by the *DocGen reranker*. Formally, we aim to maximize the following:

$$\mathbb{E}_{\boldsymbol{x}\sim\mathcal{D}, \boldsymbol{z}\sim p_{\text{PLM}}(\cdot|\boldsymbol{x}), \boldsymbol{y}\sim p_{\text{LLM}}(\cdot|\boldsymbol{x},\boldsymbol{z})}[\mathcal{R}(\boldsymbol{x},\boldsymbol{y})]. \quad (1)$$

In the three-step pipeline of DocGen, we freeze the parameters of the LLMs for the first (expanding query) and the third (generating documents) steps, our aim is to optimize the LLM for the second step (highlighting query) as a policy LM and maximize the above objective 1 by using proximal policy optimization (PPO). We use *DocGen reranker* as our reward function.

The process of generating a sequence of highlighted query tokens can be seen as a Markov decision process (MDP). In this process, there are different states, actions, rewards, and probabilities of transitioning between states. During each step, the agent chooses an action, which is a token, from the vocabulary of the LLM based on the current weight values of the LLM. The vocabulary of the LLM represents the set of possible actions for the policy LM (agent). The policy LM takes into account the input query and previous tokens to generate the next token. The process continues until an end-of-sequence token is generated, indicating the completion of the query.

## 4 Experimental Setup

**Retrieval methods.** We use a two-stage retrieval architecture (Nogueira and Cho, 2019) consisting of initial retrieval with BM25, followed by a neural reranker. The collection is indexed using pyserini.[5] We retrieve 1000 candidate documents for each query using BM25. Subsequently, we rerank the candidate documents using MonoT5, which is an adaptation of the T5 model (Raffel et al., 2020) for text ranking proposed by Nogueira et al. (2020). We fine-tune MonoT5-base (220M parameters) with a constant learning rate of $10^{-3}$ and an equal number of positive and negative examples in each batch of size 64. We did not conduct experiments with the 3B and 11B versions due to their computational cost.

**Baselines for data augmentation.** We select the state-of-the-art data augmentation method using LLMs for the passage reranking task, **InPars**, as our main baseline. In addition to the InPars, we replicate **GenRead** (Yu et al., 2023) that recently has set a new state-of-the-art for the open-domain question answering task, in which the focus is not on data augmentation for training rerankers, but the goal is to generate a relevant document for the given query and then generate the final answer from the generated document. We consider the few-shot variation of GenRead as another baseline for generating documents given the query. We use the same prompts released by the author.[6] Moreover, we replicate **Query2Doc (Q2D)** (Wang et al., 2023) that generate documents given a query in order to expand the query and improve retrieval effectiveness using the expanded query. We adapt Q2D for data augmentation by using their prompt and generating documents. We use the generated documents for building the training data.

To ensure a fair comparison, we do not compare our results with InPars-v2 (Jeronymo et al., 2023) and Promptagator (Dai et al., 2023) results. InPars-

---

[5]https://github.com/castorini/pyserini
[6]https://github.com/wyu97/GenRead

Table 1: Main results. On each dataset, we generate synthetic data given the queries from the training set for document generation methods and given the randomly sampled documents from the corpus for query generation methods. Note that on DL'20, we still utilize the training queries of MS MARCO since TREC-DL is solely an evaluation query set based on the corpus of MS MARCO. Significance is shown with † for DocGen compared to the best baseline (GenRead) and with * for DocGen-RL compared to DocGen. We use MonoT5 with 220M parameters for all the rerankers. The cut-off for nDCG and MRR is 10 and for MAP is 1000, respectively. Statistical significance was measured with a paired t-test ($p < 0.05$) with Bonferroni correction for multiple testing.

| Retriever | Data Augmentor | NQ nDCG | MS MARCO MRR | DL'20 | | HotpotQA nDCG | Fever nDCG |
|---|---|---|---|---|---|---|---|
| | | | | MAP | nDCG | | |
| **First-stage retriever** | | | | | | | |
| BM25 (replicated) | | .329 | .187 | .286 | .480 | .633 | .651 |
| **Rerankers** | | | | | | | |
| MonoT5 | InPars (Bonifacio et al., 2022) | .335 | .259 | .360 | .576 | - | - |
| MonoT5 | InPars (replicated) | .337 | .223 | .357 | .569 | .627 | .653 |
| MonoT5 | GenRead (replicated) | .368 | .230 | .354 | .570 | .629 | .668 |
| MonoT5 | Q2D (replicated) | .309 | .158 | .252 | .437 | .610 | .634 |
| **Rerankers w/ DocGen** | | | | | | | |
| MonoT5 | DocGen (Ours) | .467† | .275† | .398† | .580† | .647† | .693† |
| MonoT5 | DocGen-RL (Ours) | .517* | .332* | .421* | .618* | .663* | .720* |
| **Human Annotations** | | | | | | | |
| MonoT5 | – | .567 | .381 | .491 | .714 | .695 | .802 |

v2 uses the reranker that is trained on MS MARCO to filter the generated synthetic data; after the synthetic data filtering and generation, InPars-v2 first warms up a passage reranker using the whole training set of MS MARCO and then continue training the reranker on the generated synthetic data. However, DocGen has not used the supervision from MS MARCO for either synthetic data filtering or passage reranker training. Promptagator performs first-stage retrieval (full ranking) with customized prompts per dataset, which is different from our approach where we focus on reranking with a single prompt for all the datasets.

To have a fair comparison between data augmentation approaches in our replication, we use the same LLM (BLOOM-560M) and same filtering approach for all of the data augmentation methodologies. We then generate an equal amount of synthetic data which is $100,000$ synthetically generated queries or documents followed by (Bonifacio et al., 2022).

**Dataset and metrics.** We conduct our experiments on the MS MARCO-passage collection (Nguyen et al., 2016) and the TREC Deep Learning track (TREC-DL'20) (Craswell et al., 2021), as well as the BEIR versions of Natural Questions (NQ) (Kwiatkowski et al., 2019), HotPotQA (Yang et al., 2018), and Fever (Thorne et al., 2018) datasets. We report the official metrics for each dataset, which are NDCG@10 for BEIR's datasets (NQ, HotPotQA, and Fever), MRR@10 for MS MARCO, and NDCG@10 and MAP@1000 for TREC-DL'20. The MS MARCO-passage dataset contains about 1 million natural language queries (average length: $7.5$ words) and has been extensively used to train neural retrievers for ranking because of the large number of queries. Following prior work on MS MARCO (Khattab and Zaharia, 2020; Lin et al., 2021; MacAvaney et al., 2020; Zhuang and Zuccon, 2021; Zhuang et al., 2021), we use the dev set ($\sim 7k$ queries) for our empirical evaluation. TREC-DL'20 consists of 54 queries, and the retrieval for these queries is based on the passage corpus of MS MARCO. We follow the common practice on TREC-DL (Craswell et al., 2021, 2020) and use $nDCG@10$ and $MAP@1000$ for evaluation.

**LLM selection.** In our preliminary experiments, we examine four LLMs, namely, LLaMA-7B (Touvron et al., 2023), Alpaca-Lora (Wang, 2023), Flan-T5-xxl (Chung et al., 2022), BLOOM-560M (Scao et al., 2022). We observe that the smallest and most efficient model, BLOOM-560M, consistently generates higher-quality documents compared to the other models for our document generation task. Therefore, we select BLOOM-560M as the main

LLM for our experiments. We found that Alpaca-Lora and Flan-T5 tend to generate less informative and shorter text. LLaMA tends to produce noisy text, e.g., LaTeX Code.

**Training policy LLM with RL.** We train the policy network for $30k$ episodes, $5$ epochs per batch with a batch size of $2$ and a learning rate of $2 \times 10^{-6}$. We set the other hyperparameters to the optimal parameters that are suggested in (Li et al., 2023). If the agent (i.e., LLM) chooses an action (i.e., token), which is not a token in the given query that is going to be highlighted, we assign a penalty reward of $-0.25$, drawing inspiration from (Li et al., 2023) that gives the same penalty for finding hint keywords from a text to do summarization.

## 5 Results

In this section, we address the following research questions, assessing the effectiveness of our proposed methods, DocGen and DocGen-RL, from different perspectives:

- **RQ1:** What is the effectiveness of DocGen compared to the existing state-of-the-art baselines for data augmentation in IR models?
- **RQ2:** To what extent does employing RL improve our pipeline?
- **RQ3:** What is the impact of each step of few-shot learning and reinforcement learning on DocGen and DocGen-Rl?
- **RQ4:** To what extent does scaling LLM or MonoT5 improve the proposed method?

**Main results (RQ1 and RQ2).** Table 1 presents the results of training MonoT5 on the augmented data using DocGen and DocGen-RL, comparing it to the previous state-of-the-art few-shot learning data augmentation method, InPars, and other competitive baselines including GenRead (Yu et al., 2023). BM25 achieves competitive scores to InPars and GenRead on HotpotQA and Fever. Overall, both variations of our proposed method, DocGen and DocGen-RL, outperform all the baselines and demonstrate significant improvements in retrieval performance across multiple datasets.

**Ablation and RL analysis (RQ3).** To investigate the impact of each step of few-shot learning and reinforcement learning on DocGen and DocGen-Rl, we conduct an ablation study, of which the results are shown in Table 2, to evaluate the impact of different components and variations on the performance of DocGen and DocGen-RL models in terms of nDCG@10 on NQ-test. The first sec-

Table 2: Ablation study on DocGen and RL-training analysis on DocGen-RL using nDCG@10 for evaluation.

| Dataset | NQ-test |
|---|---|
| **Ablation study on DocGen** | |
| DocGen w/o expanding | .370 |
| DocGen w/o highlighting | .363 |
| DocGen w/o expanding & highlighting | .351 |
| DocGen | .467 |
| **RL-training analysis on DocGen-RL** | |
| DocGen + only RL on highlighting (DocGen-RL) | .517 |
| DocGen + only RL on expanding | .473 |
| DocGen + only RL on doc generation | .448 |

Table 3: Impact of scaling on DocGen. Evaluation on NQ-test in terms of nDCG@10.

| | |
|---|---|
| BLOOM-560M and T5-base (220M) | .467 |
| BLOOM-3B | .482 |
| T5-large (770M) | .495 |

tion of the table focuses on the ablation study of DocGen's few-shot learning-based pipeline. It explores the impact of removing specific components on the model's performance. The results for these settings indicate that highlighting is more important than expanding while both steps do contribute to the effectiveness of the model. The second section of the table presents the analysis study conducted on DocGen-RL. The results in second section show that RL training on highlighting can achieve the highest improvement for DocGen. Similarly, like RL training for highlighting, we keep other steps frozen when we perform RL training for expanding or document generation, and we use the same reward function. RL training for expanding queries can slightly improve effectiveness, while RL training for document generation decreases effectiveness. This could be because generating a document is a more challenging task, and training the LLM on this task using RL could be more challenging.

**Scaling impact (RQ4).** We investigate the impact of scale on DocGen from two perspectives: (i) the number of parameters of BLOOM for generation, and (ii) the number of parameters of the trained MonoT5 on the augmented data by DocGen's pipeline. To address (i), we analyze to what extent DocGen improves by increasing the number of LLM's parameters and evaluate the effectiveness of DocGen with a version of BLOOM that

Table 4: Analysis of the overlap between synthetic and realistic data on the NQ dataset. Overlap refers to the average number of words overlapping between a query and a document. Q, D, and W refer to the queries and documents and words. The term "Expanded" refers to the fact that we expand the human queries in the *DocGen pipeline*. As we remove highlighting marks from the final training data constructed by DocGen, expanding the query is the only step in the *DocGen pipeline* that has an effect on the length of the query.

| Augmentor | Query | Documents | Overlap | Avg # W in Q | Avg # W in D |
|---|---|---|---|---|---|
| InPars (Bonifacio et al., 2022) | Synthetic | Human | 6.54 | 6.78 | - |
| GenRead (Yu et al., 2023) | Human | Synthetic | 4.39 | - | 57.02 |
| DocGen | Expanded | Synthetic | 2.22 | 7.72 | 69.89 |
| DocGen-RL | Expanded | Synthetic | 2.58 | 7.72 | 71.24 |
| - | Human | Human | 3.20 | 9.15 | 79.02 |

Table 5: DocGen and DocGen-RL VS. human supervised data using nDCG@10 for evaluation.

| Dataset | TREC DL'20 |
|---|---|
| **100k generated doc before CF** | |
| DocGen | .580 |
| DocgGen-RL | .618 |
| **1M generated doc before CF** | |
| DocGen | .652 |
| DocgGen-RL | .688 |
| **Supervised (Human)** | |
| MonoT5-MS MARCO | .714 |

is about 5.5 times bigger, with three billion parameters (BLOOM-3B) instead of the version with BLOOM 560 million parameters. For (ii), we fine-tune MonoT5 with 770 million parameters, which is about 3.5 times larger, instead of the version with 200 million parameters. Table 3 shows that we achieve a significant improvement over the results by increasing the scale of either BLOOM or MonoT5 parameters.

## 6 Discussion

**Gap between synthetic and realistic data.** We investigate to what extent different data augmentation methods are close to the human data in terms of the average length of queries and documents, as well as the average number of word overlaps within the query and document. Table 4 shows that the generated documents by DocGen and DocGen-RL, and the modified expanded queries, are closer to the human data compared to the other data augmentation methods. Moreover, we observe that generating queries from documents by InPars (Bonifacio et al., 2022) leads to a high lexical overlap between the query and generated document, as the LLMs tend to select the important words from the document as the query words during query generation, which is dissimilar from human queries in that there can be cases that the relevant document has less mutual words with the query. That is why a semantical search is important. We see that InPars, which generates queries, has the highest overlap between queries and documents, and GenRead, which also generates documents similarly to DocGen, has less lexical overlap compared to InPars.

**Can generated documents replace human-annotated documents?** We investigate this question by employing DocGen and DocGen-RL to generate synthetic documents for the one million queries of the training set queries of MS MARCO. We then compare these results with training MonoT5 on the MS MARCO training set vs. synthetic data generated by DocGen and DocGen-RL. Table 5 demonstrates the promising potential of utilizing DocGen and DocGen-RL as alternatives to human-annotated datasets as we achieve higher effectiveness that is slightly less than human-annotated data by increasing the size of our synthetic dataset.

**Computational cost analysis.** Table 6 represents an analysis of the inference time between DocGen and InPars. We report two variants of InPars: (i) the original InPars (Bonifacio et al., 2022) implementation which adopts GPT-J-6B, and (ii) a more efficient variant which adopts BLOOM-560M. We observe that both DocGen and DocGen-RL are over twice as efficient as InPars-GPT in inference (107 seconds vs. 237 seconds). However, they are less efficient than InPars-BLOOM (72 seconds vs. 107 seconds). We believe that the additional computational costs can be justified

Table 6: Efficiency analysis of different data augmentation methods for information retrieval.

| Data Augmentor | LLM | Latency | nDCG@10 on NQ |
|---|---|---|---|
| InPars (Bonifacio et al., 2022) | GPT-J-6B | 237 seconds | 0.335 |
| DocGen | BLOOM-560M | 107 seconds | 0.467 |
| DocGen-RL | BLOOM-560M | 107 seconds | 0.517 |
| InPars (replicated) | BLOOM-560M | 72 seconds | 0.337 |

Table 7: Impact of DocGen and DocGen-RL on building training dataset for further fine-tuning MonoT5-MSMARCO vs. fine-tuning MonoT5 from scratch.

| Data Augmentor | Retriever | nDCG@10 |
|---|---|---|
| DocGen | MonoT5 | 0.467 |
| DocGen | MonoT5-MSMARCO | 0.492 |
| DocGen-RL | MonoT5 | 0.517 |
| DocGen-RL | MonoT5-MSMARCO | 0.539 |

by the large performance gain that both models achieve (nDCG@10 on NQ: 0.517 (DocGen-RL) vs. 0.337 (InPars-BLOOM)).

**Impact of DocGen on a trained retriever.** To analyze the impact of DocGen on the effectiveness of the already trained retriever model, we fine-tuned starting from the checkpoint of MonoT5-MSMARCO with 220M parameters[7], pre-trained on MS MARCO, and analyze to what extent further fine-tuning it on the augmented data will have an impact on its effectiveness. Table 7 shows that DocGen can make improvements in this scenario as well while this impact is not quite large. This observation is in line with InPars V2 ((Jeronymo et al., 2023)) which shows that further fine-tuning from MonoT5-MSMARCO does not lead to a large improvement. This is because further fine-tuning a well-trained MonoT5 could diverge the retriever from an optimal weight.

## 7 Conclusions

This paper presents a new perspective for constructing training datasets in information retrieval without human labels. It proposes two novel query-to-document generators, called DocGen and DocGen-RL. The motivation behind this approach is to overcome the limitations of existing methods that generate queries from documents. Additionally, we automatically enhance low-quality queries be-

fore generating documents. Extensive experiments demonstrate that both DocGen and DocGen-RL significantly outperform existing approaches, including InPars. This work showcases the potential of leveraging LLMs for document generation and highlights the significance of document augmentation in information retrieval. We emphasize the importance of improving the quality of training datasets. The findings of this study contribute to advancing the state-of-the-art in IR and provide valuable insights for future research in this domain.

## Limitations

We showed in this paper that generative LLMs can create data to train effective retrieval models. We should stress that other aspects of evaluation have not been investigated in this paper, specifically the effect of biased information in the generated documents on biases in the document ranking. It is well-known that LLMs have biases reflected in their output. In this work we do not study the effect of such biases on the performance of the reranker, potentially biasing the retrieved documents. Another problem of generative LLMs is that the factuality of the output cannot be guaranteed. Even though factually incorrect information in the generated data (as a result of LLM's hallucination) is not likely to be harmful in the IR context, because only information that is truly contained in the document collection can be retrieved by a retrieval model. We do not systematically study and quantify the effect of hallucinated data on the performance of

---

[7]urlhttps://huggingface.co/castorini/monot5-base-msmarco

the ranker.

## Ethics Statement

Reiterating the importance of understanding and quantifying LLMs bias in generating data, we believe that this can lead to unforeseen ethical consequences. Therefore, we need to carefully study potential biases that exist in the data and formalize their impact on the generated data and the trained model. While in this work we demonstrate the potential of LLMs in generating synthetic data to improve IR systems, we believe that such approaches should be applied with great care in real-world search systems and the system designers should take into account the existence of such biases.

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

# A  Appendix

## A.1  Prompts

Figure 3, 4, and 5 show the prompts for expanding the orignal query, highlighint the expanded query, and document generation given expanded and highlighted query.

## A.2  Safeguard Data Leakage

In order to being sure that there is no data leakage between the evaluation datasets in our experiments and the training set of the BLOOM, we assess in the BLOOM training data documentation (Laurençon

Figure 3: Query expansion with in-context-reasoning prompt.

Figure 4: Query highlighting with in-context-reasoning prompt.

Example1:
Query: What is the recommended amount of [caffeine] intake during [pregnancy], and are there any potential risks associated with consuming small amounts of [caffeine] while [pregnant]?
Relevant Document: We don't know a lot about the effects of caffeine during pregnancy on you and your baby. So it's best to limit the amount you get each day. If you are pregnant, limit caffeine to 200 milligrams each day. This is about the amount in 1½ 8-ounce cups of coffee or one 12-ounce cup of coffee.

Example 2:
Query: Which [fruit] is exclusive to [Australia] and provide some additional details about it?
Relevant Document: Passiflora herbertiana. A rare passion fruit native to Australia. Fruits are green-skinned, white fleshed, with an unknown edible rating. Some sources list the fruit as edible, sweet and tasty, while others list the fruits as being bitter and inedible.assiflora herbertiana. A rare passion fruit native to Australia. Fruits are green-skinned, white fleshed, with an unknown edible rating. Some sources list the fruit as edible, sweet and tasty, while others list the fruits as being bitter and inedible.

Example 3:
Query: What is the size of the [canadian military] ahd what is the number of active personnel and reserve members?
Relevant Document: The Canadian Armed Forces. 1 The first large-scale Canadian peacekeeping mission started in Egypt on November 24, 1956. 2 There are approximately 65,000 Regular Force and 25,000 reservist members in the Canadian military. 3 In Canada, August 9 is designated as National Peacekeepers' Day.

Example 4:
Query: {query_text}
Relevant Document:

Figure 5: Document generation with in-context-reasoning prompt.

et al., 2022) if any of the BEIR collection datasets or MSMARCO dataset are involved in the training dataset of the BLOOM. We made sure that there is no mutual dataset within BLOOM training data, BigScience Corpus (Laurençon et al., 2022), and our evaluation datasets.

## A.3 Optimal character for highlighting.

To determine the most effective highlighting character, we compare various characters, including less than and greater than symbols ('<>'), asterisks ('**'), square brackets('[]'), parentheses ('()'), and curly braces ('{}'). We evaluate these highlighting characters based on their impact on the quality of training data by assessing the performance of MonoT5 fine-tuned on augmented data for NQ queries using each highlighting character. Table 8 demonstrates that using parentheses and square brackets as highlighting characters leads to improvement while using other highlighting characters can even lead to a drop in effectiveness. We think this is because probably those symbols do not trigger any part of the memory of the LLM and act as noise. The square bracket that is used by DocGen is shown to be an optimal highlighting character.

Table 8: Impact of highlighting character on DocGen. Evaluation on NQ-test in terms of nDCG@10.

| Highlighting character | |
|---|---|
| '<>' (e.g., what are <calling cards>?) | .301 |
| '{}' (e.g., what are {calling cards}?) | .319 |
| '**' (e.g., what are *calling cards*?) | .360 |
| '()' (e.g., what are (calling cards)?) | .391 |
| '[]' (i.e., what are [calling cards]?) | .467 |
| DocGen w/o highlighting | .363 |

Example 4:
Query: what goes on the bottom of shepherd's pie
Relevant document: \begin{itemize}
 \item 50g butter
        \begin {itemize}\item melted
        \end{ulis}t
        \n\item flour
        \uline{15g} \
        \linebreak
        \lipsum[1]
        \hfill
        \vspace{2pt}
        \textbf{
                \textit{ }
                \textbullet
                \par
                \noindent
        \bigskip
 \newline
  \hline
*
**\end {enumerate}*\n  *\end* \\\\\ \\
 &&&\n\\\\ \ \\ \ \  \\ \\  & \ & & \\ &  &\n \\\\  &&&& \& \&\&\\& &\\\n \
 &&&&\&\
 &&& \\& &\\ \\&&&&

Figure 6: Example of a noisy generated document by LLaMA.

## A.4 Example of a noisy generated document by LLaMA

Figure 6 provides an example of a generated document by LLaMA that has a large amount of noisy text. This could show an example of why LLaMA is not suitable for our task. We observed that for more than half of the queries of Natural Question (NQ) dataset, LLaMA produces noisy text.