# OpenReview forum: "Expand, Highlight, Generate: RL-driven Document Generation for Passage Reranking"
_EMNLP/2023/Conference — EMNLP 2023 Main_

### Official Review · Reviewer_qmXW · 2023-08-05

**Soundness:** 4

**Excitement:**

4: Strong: This paper deepens the understanding of some phenomenon or lowers the barriers to an existing research direction.

**Missing References:**

A very recent paper (ACL23) that could be related to this. They designed data augmentation techniques to improve a generative QA model:
- Learning Answer Generation using Supervision from Automatic Question Answering Evaluators

**Paper Topic And Main Contributions:**

This paper presents a novel technique to perform data augmentation named DocGen. This technique consists of a pipeline organized in three sequential steps. In the first step, named "query expansion," short keyword-based queries are expanded to longer queries in natural language to maximize the effectiveness of LLM generation. In the second step, named "query highlighting", they use a LLM to highlight relevant words in the expanded query (e.g., "Is a duck bigger than an elephant?" -> "Is a [duck] [bigger than] an [elephant]?") to explicitly manipulate the document generator to pay more attention to those terms. In the third and last step, the expanded and highlighted query is used as a prompt to generate a relevant document. To further refine the quality of the generated documents, they apply a consistency filter based on the output of a ranker (keeping the best document only).

They measured the performance of their data augmentation approach on several benchmarks showing that DocGen and DocGen-RL (a finetuned version of DocGen) allow training stronger rankers to outperform existing data augmentation approaches.

To conclude, the main novelty presented in this paper regards the "query highlighting" step, which shows that the generation can be easily improved, leading to a better generation pipeline and, consequently, better results on the downstream tasks.

**Questions For The Authors:**

- The paper mentions that BLOOM-550M was the best model among the considered ones. Could the other models have lower performance than BLOOM, thanks to the implicit highlighting method learned from it during its training (lines 265-267)?
- How many times the top-1 retrieved document (mentioned in lines 302-304) is the generated one?

**Reasons To Accept:**

- The presented results show a significant performance improvement.
- The methodology and the experiments are easy to reproduce.

**Reasons To Reject:**

- Although the experimental setting is well defined and sufficient, the paper could benefit from (i) results considering different models and approaches (e.g., LLama for the generation, DeBERTa for the ranker, and DPR for the retrieval), (ii) a qualitative analysis of the generated documents o support Table 4 and the Discussion Section (e.g., by adding some example of good and bad generations).
- The paper presentation could be improved (with a minor review) since it is not always easy to follow, and sometimes it is verbose (see the suggestions).

**Reproducibility:**

4: Could mostly reproduce the results, but there may be some variation because of sample variance or minor variations in their interpretation of the protocol or method.

**Reviewer Confidence:**

4: Quite sure. I tried to check the important points carefully. It's unlikely, though conceivable, that I missed something that should affect my ratings.

**Typos Grammar Style And Presentation Improvements:**

- Check the typos:
line 219 -> an a consistency
line 246 -> documents 2 (I think it should be "Fig. 2”)
line 340 -> few-show
line 358 -> and the

- Highlight the best results in the tables to improve the readability.

- Repeated concepts could lead the paper to be verbose (e.g., GenRead has been described in both the experimental and related work sections).

---

> ### Author Rebuttal · Authors · 2023-08-29
>
> *Question A*
>
> > The paper mentions that BLOOM-550M was the best model among the considered ones. Could the other models have lower performance than BLOOM, thanks to the implicit highlighting method learned from it during its training (lines 265-267)?
>
> Based on our preliminary experiment on four LLMs, namely, LLaMA-7B (Tou-vron et al., 2023), Alpaca-Lora (Wang, 2023), Flan-T5-xxl (Chung et al., 2022), and BLOOM-560M (Scao et al., 2022), all of the four LLMs could indeed take advantage of highlighting and the impact of highlighting is not only limited to BLOOM-560M.
>
> It is noteworthy that in our preliminary experiments, we observed that LLaMA tends to produce noisy text (e.g., LaTeX code) for the queries that we used, and Alpaca-Lora tends to produce very short documents. We will add examples of such cases in the appendix of the paper's camera-ready version. You can also view an example of this case for LLaMA at the end of our rebuttal response.
>
> ---
>
> *Question B*
>
> > How many times the top-1 retrieved document (mentioned in lines 302-304) is the generated one?
>
> We analyzed this for all of the used datasets in this work. Our analysis reveals that, on average across the datasets, the top-1 retrieved document corresponds to the synthetically generated document in 52\% of instances for DocGen-RL and 44\% of instances for DocGen.
>
>
> ---
>
> *Reason to reject A*
>
> > Although the experimental setting is well defined and sufficient, the paper could benefit from (i) results considering different models and approaches (e.g., LLama for the generation, DeBERTa for the ranker, and DPR for the retrieval), (ii) a qualitative analysis of the generated documents o support Table 4 and the Discussion Section (e.g., by adding some example of good and bad generations).
>
> Regarding **(i)**, we have tried LLaMA for the generation but it tended to produce noisy text (with LaTeX code) for more than half of the questions. We also tried Alpaca-Lora but it tended to produce very short documents. We will add examples of such cases in the appendix of our paper's camera-ready version. We also added an example of noisy text produced by LLaMA at the end of this rebuttal.
>
> For the sake of limiting the scope of our work, we selected reranking with MonoT5 following InPars as a state-of-the-art ranker. We believe experimenting with DeBERTa and DPR would be possible as our method is a data augmentation methodology that can potentially be beneficial to any retrievers.  We will do this experiment in future work.
>
> Regarding **(ii)**, thanks for suggesting this insightful analysis. We will add this in the camera-ready version as a qualitative analysis in order to support Table 4. We will show how generating queries from documents will lead to more mutual words than generating documents from queries.
>
> ---
>
> *Reason to reject B*
>
> > The paper presentation could be improved (with a minor review) since it is not always easy to follow, and sometimes it is verbose (see the suggestions).
>
> Thank you for the suggestion. We have addressed all the "Grammar Style And Presentation Improvements" suggested by you and other reviewers in the camera-ready version.
>
>
> ---
>
> ## Example of a noisy generated document by LLaMA
>
> In the following, we provide an example of a generated document by LLaMA that has a large amount of noisy text. This could show an example of why LLaMA is not suitable for our task. We observed that for more than half of the queries of NQ, LLaMA produces noisy text.
>
>
> ```Example 4:
> Query: what goes on the bottom of shepherd's pie
> Relevant document: \begin{itemize}
>   \item 50g butter
> 	\begin {itemize}\item melted
>   	\end{ulis}t
>  	\n\item flour
>    	\uline{15g} \
>     	\linebreak
>      	\lipsum[1]
>       	\hfill
>        	\vspace{2pt}
>         	\textbf{
>           	\textit{  }
>            	\textbullet
>             	\par
>              	\noindent
>          	\bigskip
>  \newline
>    \hline
> *
> **\end {enumerate}*\n  *\end* \\\\\\ \\
>  &&&\n\\\\ \ \\ \ \  \\ \\  & \ & & \\ &  &\n \\\\  &&&& \& \&\&\\& &\\\n \ &&&&\&\
> &&& \\& &\\ \\&&&&```

---

### Official Review · Reviewer_y3cg · 2023-08-05

**Soundness:** 4

**Excitement:**

3: Ambivalent: It has merits (e.g., it reports state-of-the-art results, the idea is nice), but there are key weaknesses (e.g., it describes incremental work), and it can significantly benefit from another round of revision. However, I won't object to accepting it if my co-reviewers champion it.

**Paper Topic And Main Contributions:**

* The paper is about improving neural based document retrieval, focusing on generated synthetic documents.
* The main contribution of this work is 1) its investigation of how to generate good synthetic documents given queries, and 2) its RL training to determine query highlights that can help LLMs generate better documents.

**Questions For The Authors:**

* how to determine important words in square brackets in the prompt examples? Did you find any patterns of LLM output what do they prefer to highlight, before and after RL training?

* The authors follow Promptagator to conduct consistency filtering. Any reason why only the top-1 is selected? Is there any results on Top-K analysis? The w/ and w/o CF parts are missing in the paper.

**Reasons To Accept:**

* The paper shows good results comparing to the common (document —> query) path, which showed a potential data augmentation direction for document retrieval.
* The paper shows easy-to-follow ablation study to understand most parts of its design such as w/o query expansion,  w/o highlighting, w/o RL.

**Reasons To Reject:**

* Is it possible that the performance gap in Table 1 is coming from the instruction following ability of different LLMs? For example, BLOOM-560M has better ability to generate documents as the authors indicated in line 476, however, it is not good and generate queries so the query-generator baselines have more disadvantages. Why ChatGPT is not considered to be used as the base generator?

* The results are compared fully without using the supervision data (and that’s why InPars-V2 and Promptagator are not in the comparison as the authors justified). However, it will be great to see how does this document augmentation can help on top of the existing supervised retriever, since we already have those training signals and the trained retrievers can be used directly OOTB for other domains. So it makes more sense to show that no matter with augmented queries or augmented documents (or both together), how can it add additional performance boost to the final retriever model.

**Reproducibility:**

4: Could mostly reproduce the results, but there may be some variation because of sample variance or minor variations in their interpretation of the protocol or method.

**Reviewer Confidence:**

3: Pretty sure, but there's a chance I missed something. Although I have a good feel for this area in general, I did not carefully check the paper's details, e.g., the math, experimental design, or novelty.

---

> ### Author Rebuttal · Authors · 2023-08-29
>
> *Question A*
>
> > how to determine important words in square brackets in the prompt examples? Did you find any patterns of LLM output what do they prefer to highlight, before and after RL training?
>
> Regarding determining important words in the examples, we empirically determined that highlighting words around the main subject of the query is beneficial and also feels natural. Therefore, we chose to highlight tokens with significant roles in the prompt examples.
>
> We analyze the highlighting pattern before and after RL training and report the number of highlighted tokens based on their linguistic roles using the NLTK library.
>
> | Highlighted word POS | Before RL                           | After RL      |
> |-----------------------|-------------------------------------|---------------|
> | Pronouns              | 44,993 words                        | 30,040 words  |
> | Nouns                 | 120,665 words                       | 151,236 words |
> | Entities              | 87,942 words                        | 120,160 words |
> | Verbs                 | 38,566 words                        | 39,851 words  |
> | Non-highlighted words | 981,811 words                       | 930,336 words |
> | Total number of words:           1,281,795 words
>
>
> As shown in the above table, we find that the number of highlighted pronouns is reduced after RL training, while the number of highlighted nouns and entities increases. However, a considerable number of highlighted pronouns still remains highlighted after RL training. Furthermore, we observe that the number of highlighted verbs remains nearly the same.
>
> ---
>
> *Question B*
>
> > The authors follow Promptagator to conduct consistency filtering. Any reason why only the top-1 is selected? Is there any results on Top-K analysis? The w/ and w/o CF parts are missing in the paper.
>
> Thank you for the constructive suggestion. In the following table, we report analysis on considering top-k for consistency filtering, ranging from 1 to 5, where we report the performance of DocGen-RL w/ and w/o Consistency Filtering (CF) on the Natural Question dataset.
> The following table demonstrates that including synthetic documents ranked at lower positions within the training data leads to reduced effectiveness. This observation aligns with prior research, which advocated for selecting the top-ranked document (k=1) to ensure training data quality.
>
> | Top-k | nDCG@10   |
> |-------|--------|
> | 1     | 0.5175 |
> | 2     | 0.5009 |
> | 3     | 0.4887 |
> | 4     | 0.4401 |
> | 5     | 0.4026 |
>
>
> Regarding **"The w/ and w/o CF parts are missing in the paper."**, we will add this experiment to our result table for the camera-ready version of the paper. As an example, we report the below table that showcases the effectiveness of DocGen, both with and without CF on the Natural Question dataset in terms of nDCG@10. As expected, both models experience a drop in effectiveness without CF, underscoring the role of consistency filtering in maintaining the quality of synthetically generated data.
>
> | Data Augmentor | Retriever | nDCG@10 |
> |----------------|-----------|---------|
> | DocGen w/o CF  | MonoT5    | 0.375   |
> | DocGen w/ CF   | MonoT5    | 0.467   |
>
> ---
>
> *Reason to reject A*
>
> > Is it possible that the performance gap in Table 1 is coming from the instruction following ability of different LLMs? For example, BLOOM-560M has better ability to generate documents as the authors indicated in line 476, however, it is not good and generate queries so the query-generator baselines have more disadvantages.
>
> First, let us clarify that we do not claim that BLOOM-560M is better at generating documents than generating queries; in line 476 we say that BLOOM-560M has better document-generating capabilities among the four LLMs that we investigated (LLaMA-7B, Alpaca-Lora, Flan-T5-xxl, BLOOM-560M). To prevent unfair advantages over query generation approaches, we compared our method to the original implementation of InPars, which uses an effective model for query generation (Table 1 of the paper, row 2, InPars (Bonifacio et al., 2022)). Moreover, our methods achieve better effectiveness with using BLOOM-560M which is 12 times smaller than GPT-J-6B used by InPars in its original results.
>
> > Why ChatGPT is not considered to be used as the base generator?
>
> There are two reasons:
>
> - **Reliability.** We do not consider the closed-source models such as ChatGPT and GPT 3.5/4.0 to ensure the reliability of our evaluation because it is unclear if there is a data leakage between the pre-training data of these closed-source models and the datasets that we need to use for evaluation.
>
> - **Reproducibility.** We would like to ensure the reproducibility of our paper. ChatGPT and GPT 3.5/4.0 keep updating, and it is hard for researchers to fully replicate their results.
>
>
> ---
>
> *Reason to reject B*
>
> > The results are compared fully without using the supervision data (and that’s why InPars-V2 and Promptagator are not in the comparison as the authors justified). However, it will be great to see how does this document augmentation can help on top of the existing supervised retriever, since we already have those training signals and the trained retrievers can be used directly OOTB for other domains. So it makes more sense to show that no matter with augmented queries or augmented documents (or both together), how can it add additional performance boost to the final retriever model.
>
> Thank you for your advice. We investigated this based on your feedback.
>
> To analyze the impact of DocGen on the effectiveness of the final retriever model, we fine-tuned starting from the checkpoint of MonoT5-MSMARCO with 220M parameters, pre-trained on MS MARCO, and analyze to what extent further fine-tuning it on the augmented data will have an impact on its effectiveness.
>
> | Data Augmentor | Retriever      | nDCG@10 |
> |----------------|----------------|---------|
> | DocGen         | MonoT5         | 0.467   |
> | DocGen         | MonoT5-MSMARCO | 0.492   |
> | DocGen-RL      | MonoT5         | 0.517   |
> | DocGen-RL      | MonoT5-MSMARCO | 0.539   |
>
>
> The above table shows that DocGen can make improvements in this scenario as well while this impact is not quite large. This observation is in line with InPars V2 (Jeronymo et al.,2023) which shows that further fine-tuning from MonoT5-MSMARCO does not lead to a large improvement. This is because further fine-tuning a well-trained MonoT5 could diverge the retriever from an optimal weight.

---

### Official Review · Reviewer_yxMC · 2023-08-17

**Soundness:** 4

**Excitement:**

3: Ambivalent: It has merits (e.g., it reports state-of-the-art results, the idea is nice), but there are key weaknesses (e.g., it describes incremental work), and it can significantly benefit from another round of revision. However, I won't object to accepting it if my co-reviewers champion it.

**Paper Topic And Main Contributions:**

This work explores a new paradigm to incorporate large language models (LLMs) for data augmentation to improve the passage reranking in information retrieval (IR) tasks by generating synthetic relevant passages given a query, while previous studies focus on generating synthetic queries given a document. Along with their exploration, they propose the approach DocGen, which applies query expansion and keyword highlighting to trigger the LLMs to generate better relevant documents, and DocGen-RL, a reinforcement learning (RL) based method to improve further the keyword highlighting process by updating LLM’s parameters based on a relevance reward from a trained T5-based retriever. Extensive experiments and analyses are conducted to address the following questions: (1) whether their method outperforms unsupervised baselines. (2) which components contribute to the improved performance? (3) whether scaling LLMs can improve the performance. (4) What is the gap between synthetic and realistic data?

The contribution lies in the proposed new data augmentation paradigm, a.k.a. generating document given a user query, the proposed three-step pipeline, named DocGen, and RL-based method, named DocGen-RL, which both outperform existing unsupervised baselines and achieves a promising performance based on this data augmentation paradigm. Extensive experiments and analysis shed light on future research.

**Questions For The Authors:**

A. Could the authors elaborate on whether they use expended queries during the inference process for BM25 and MonoT5 models?

B. What is the motivation to apply LLMs, instead of existing query expansion methods, for query expansion?

C. It is overclaimed to say that the proposed method leverages the maximum capacity of LLMs in generating synthetic data. (Line 27)

D. It may be better to have the ablation study about the number of examples used in the prompt.


6. Claveau, V. Neural text generation for query expansion in information retrieval. IEEEWICACM Int. Conf. Web Intell. 202–209 (2021) doi:10.1145/3486622.3493957.

**Reasons To Accept:**

It is a reasonable paradigm to incorporate LLMs to generate contextual documents given a query, which aligns with the research on autoregressive information retrieval like [1,2,3,4] and corresponding applications like knowledge-intensive tasks[5], although this work applies this idea to data augmentation phase.

Previous studies focus on feeding a document to LLMs to generate a query. However, it is not trivial to generate a query relevant to multiple documents due to the token limitation of LLMs. This proposed paradigm can solve this issue.

The proposed method outperforms other unsupervised baseline methods like InPars, GenRead, and Q2D on the MS MARCO, NQ, HotpotQA, Fever, and TREC-DL'20 datasets with the help of query expansion, keyword highlighting, and reinforcement learning.

Extensive analysis has been conducted about the main experiments, ablation studies, and the gap between synthetic and realistic data.

1. Cao, N. D., Izacard, G., Riedel, S. & Petroni, F. Autoregressive Entity Retrieval. Arxiv (2020) doi:10.48550/arxiv.2010.00904.

2. Tay, Y. et al. Transformer Memory as a Differentiable Search Index. Arxiv (2022) doi:10.48550/arxiv.2202.06991.

3. Lai, T. M., Ji, H. & Zhai, C. Improving Candidate Retrieval with Entity Profile Generation for Wikidata Entity Linking. Arxiv (2022) doi:10.48550/arxiv.2202.13404.

4. Bevilacqua, M., Ottaviano, G., Lewis, P., Yih, S., Riedel, S., & Petroni, F. (2022). Autoregressive search engines: Generating substrings as document identifiers. Advances in Neural Information Processing Systems, 35, 31668-31683.

5. Yu, W., Iter, D., Wang, S., Xu, Y., Ju, M., Sanyal, S., ... & Jiang, M. (2022). Generate rather than retrieve: Large language models are strong context generators. arXiv preprint arXiv:2209.10063.

**Reasons To Reject:**

A. This work assumes they have abundant user queries, which can be used to generate query-document pairs for data augmentation to handle the data sparsity issue. However, in a data sparsity situation, it may be more natural to have a certain number of documents without the corresponding user queries instead of the opposite. It is unclear in which situation we will have abundant, like 10k, user queries before we have abundant query-document pairs.

B. It is not solid enough to skip the baseline model Promptagator, even if it performs first-stage retrieval with customized prompts per dataset. For example, the authors can compare the end-to-end performance of Promptagator and their method, both with customized prompts per dataset.

C. (Minor weakness): Compared with previous LLMs-based studies, the proposed DocGen-RL requires training based on reinforcement learning, costing lots of computational resources.

**Reproducibility:**

4: Could mostly reproduce the results, but there may be some variation because of sample variance or minor variations in their interpretation of the protocol or method.

**Reviewer Confidence:**

4: Quite sure. I tried to check the important points carefully. It's unlikely, though conceivable, that I missed something that should affect my ratings.

**Typos Grammar Style And Presentation Improvements:**

Typo in Line 246: the quality of the generated documents 2. The number 2 here is meaningless.

It may be better to mention that CF refers to Consistency Filtering in Table 5.

It may not be necessary to discuss the 3.1.2 Consistency Filtering section in detail since this method is almost the same as in [7].

It may be better to remove Figure 1 or add detailed components to it.


7. PROMPTAGATOR : FEW-SHOT DENSE RETRIEVAL FROM 8 EXAMPLES.

---

> ### Author Rebuttal · Authors · 2023-08-29
>
> *Question A*
> > Could the authors elaborate on whether they use expended queries during the inference process for BM25 and MonoT5 models?
>
> No, we did not use expanded queries as the input for any of the retrievers during inference. In our paper, query expansion is only used for data augmentation, in which we expand raw queries so as to better trigger the knowledge stored in an LLM and, as a result, generate higher-quality documents.
>
> ---
>
> *Question B*
>
> > What is the motivation to apply LLMs, instead of existing query expansion methods, for query expansion?
>
> There are two reasons why we consider LLM-based query expansion: (i) we consider LLM-based query expansion since LLMs are able to conduct query expansion without the need for large amount of training data, pre-defined rules, or fine-tuning; DocGen only uses three demonstration examples to bootstrap the LLM; and (ii) there is a fundamental difference in the query expansion performed by DocGen compared to existing methods. Our objective is to rewrite the query and make it closer to natural language (rather than keyword-based), so the LLM can respond to it effectively. This objective is different than the traditional query expansion where the goal is expanding the query with additional words to enhance the likelihood of retrieving relevant documents given the expanded query. Our preliminary experiments revealed that LLMs respond more effectively to natural language queries.
>
> ---
>
> *Question C*
>
> > It is overclaimed to say that the proposed method leverages the maximum capacity of LLMs in generating synthetic data. (Line 27)
>
> Thank you for pointing this out. We will change this claim to "... our new perspective aims to utilize the capacity of LLMs in generating synthetic data **more effectively**" in the camera-ready version.
>
> ---
>
> *Question D*
>
> > It may be better to have the ablation study about the number of examples used in the prompt. Claveau, V. Neural text generation for query expansion in information retrieval. IEEEWICACM Int. Conf. Web Intell. 202–209 (2021) doi:10.1145/3486622.3493957.
>
> Thanks for your suggestion. Our choice of 3 examples was motivated by the related work (Bonifacio et al., 2022) and our preliminary experiments using different numbers of examples. We will add an ablation study to the camera-ready version of the paper on the impact of different numbers of examples on the DocGen and DocGen-RL compared to the InPars.
>
> ---
>
> *Reason to reject A*
>
> > This work assumes they have abundant user queries, which can be used to generate query-document pairs for data augmentation to handle the data sparsity issue. However, in a data sparsity situation, it may be more natural to have a certain number of documents without the corresponding user queries instead of the opposite. It is unclear in which situation we will have abundant, like 10k, user queries before we have abundant query-document pairs.
>
> An example of a situation in which our approach would be useful is in commercial search engines where there is no sparsity in terms of the number of submitted queries. We summarize the advantages of our approach in this scenario in the following:
>
> 1. **For complex queries.** For certain queries, the relevant information may be spread across multiple documents, and by using LLMs for synthetic document generation, the reranker could be optimized to provide the most optimal response that consists of information from several documents.
>
> 2. **For new or trending queries.** The search engine service provider can optimize its retriever based on the recently received and repeated (trending) queries by using DocGen to generate synthetic documents as training data given those queries, for which there are no available relevance judgments.
>
> 3. **Complementary perspective.** Our proposed methods can also be viewed as a complementary approach to query generation methodologies. In a real-world pipeline for data augmentation, both approaches may be used.
>
>
> ---
>
> *Reason to reject B*
>
> >  It is not solid enough to skip the baseline model Promptagator, even if it performs first-stage retrieval with customized prompts per dataset. For example, the authors can compare the end-to-end performance of Promptagator and their method, both with customized prompts per dataset.
>
> We replicate Promptagator using customized prompts for training rerankers on the Natural Question dataset and report its effectiveness compared to InPars, and our proposed DocGen and DocGen-RL in the following table.
>
> | Data augmentor | nDCG@10  | Retriever | LLM  |
> |----------------|-------|-----------|------------|
> | Promptagator   | 0.341 | MonoT5    | Bloom-560M |
> | InPars         | 0.335 | MonoT5    | Bloom-560M |
> | DocGen         | 0.467 | MonoT5    | Bloom-560M |
> | DocGen-RL      | 0.517 | MonoT5    | Bloom-560M |
>
>
> The results show that DocGen and DocGen-RL (both w/o customized prompt) still lead to more effective rerankers than Promptagator (w/ customized prompt). In this work, we focus on reranking and leave the analysis of the impact of synthetic document generation on training first-stage retrievers for future work.
>
>
> ---
>
> *Reason to reject C*
>
> > (Minor weakness): Compared with previous LLMs-based studies, the proposed DocGen-RL requires training based on reinforcement learning, costing lots of computational
>
> Compared to DocGen, the additional RL training for DocGen-RL can be done offline and does not need a large amount of computational resources. It only takes less than 18 hours with a single Nvidia P6000 GPU to do the RL training. This can be considered relatively low, especially when compared with the training cost of large language models. Moreover, it is noteworthy that the inference times of DocGen and DocGen-RL are equal, since they both have the same architecture, while DocGen-RL is significantly more effective.

---

### Official Review · Reviewer_LWa7 · 2023-08-20

**Soundness:** 4

**Excitement:**

4: Strong: This paper deepens the understanding of some phenomenon or lowers the barriers to an existing research direction.

**Paper Topic And Main Contributions:**

The paper presents two query-to-document generators, DocGen and DocGen-RL, with the motivation to improve existing methods generating queries from documents. Contributions are in both (1) raw queries enrichment including query expansion and query highlighting, and (2) leveraging few shots Reinforcement Learning to enhance query and generated document relevance (which is claimed to be the first work employing RL on LLMs in the field of IR data generation). To present the effectiveness of the devised pipelines and showcase the contribution of each improvement added, the authors provide analysis and experiments results with extensive benchmarking data sets, the analysis includes performance lift of the proposed methods over SOTA baselines, ablation studies on DocGen and RL-training analysis on DocGen-RL and scaling impact. The authors also provide quantitative comparison between the synthetic and realistic data.

**Questions For The Authors:**

A. In section 4 (Experimental setup) ln 431-432 it is mentioned “DocGen has not used the supervision from MS MARCO for either synthetic data passage reranker training”. I wonder if there’s anything preventing utilizing the available supervision for filtering or warming up?
B. RL training can be unstable. I wonder if the authors investigated the stability of  the trained policy network and its robustness, as well as the variance of results from DocGen-RL?
C. How does the computational cost of DocGen and DocGen-RL compare to the one of the previous methodology? Specifically I wonder how much additional computational cost are brought by adopting RL policy.

**Reasons To Accept:**

- The idea of combining query highlighting and RL is novel. The effectiveness of the RL-guided document generation is showcased in the ablation section, and in the main results section the DocGen-RL also shows stat-sig advantage DocGen standalone. The results in the paper indicate employing RL on LLMs in IR data generation can be a promising direction.
- The paper is clear-written with comprehensive analysis and experimental results provided for the readers to better understand the contributions of each component in the devised methodology.

**Reasons To Reject:**

There is no obvious concern from the reviewer.

**Reproducibility:**

4: Could mostly reproduce the results, but there may be some variation because of sample variance or minor variations in their interpretation of the protocol or method.

**Reviewer Confidence:**

3: Pretty sure, but there's a chance I missed something. Although I have a good feel for this area in general, I did not carefully check the paper's details, e.g., the math, experimental design, or novelty.

---

> ### Author Rebuttal · Authors · 2023-08-29
>
> *Question A*
> > In section 4 (Experimental setup) ln 431-432 it is mentioned “DocGen has not used the supervision from MS MARCO for either synthetic data passage reranker training”. I wonder if there’s anything preventing utilizing the available supervision for filtering or warming up?
>
> We believe effective external supervision is not consistently available across different domains or languages. This is why we focus on proposing an approach that is not dependent on external supervision and only relies on a large language model to generate synthetic documents. For example, it is more practical to apply our approach to languages other than English where large language models for those languages can be trained, but other supervision signals such as document ranking can be scarce.
>
> ---
> *Question B*
> > RL training can be unstable. I wonder if the authors investigated the stability of the trained policy network and its robustness, as well as the variance of results from DocGen-RL?
>
> Thank you for your comment. We agree that testing the robustness of the model is important. We implemented the RL part of DocGen-RL based on RL4LMs, an RL library to fine-tune language models based on human preferences. RL4LMs uses an adaptive KL controller to reduce the instability of RL training, hence we expect the model to be stable. To verify this, we tested the stability and variance of the DocGen-RL's performance by varying the random seed on the Natural Question (NQ) dataset. As shown in the following table, DocGen-RL's effectiveness has low sensitivity to the randomness brought by random seeds. We plan to extend this experiment to all the datasets and add the results to the camera-ready version of the paper.
>
> | Data Augmentor | Retriever    | nDCG@10 | nDCG@10 | nDCG@10 | nDCG@10 | nDCG@10 | nDCG@10| nDCG@10  |
> |-----------|-------------------|--------|--------|--------|--------|--------|--------|--------|
> |  |  | Seed 1 | Seed 2 | Seed 3 | Seed 4 | Seed 5 | Avg. | Std. |
> | DocGen-RL  (Ours) |  MonoT5 | 0.5175 | 0.5171 | 0.5189 | 0.5250 | 0.5177 | 0.5192 | 0.0032|
>
> ---
>
> *Question C*
> > How does the computational cost of DocGen and DocGen-RL compare to the one of the previous methodology? Specifically, I wonder how much additional computational cost is brought by adopting RL policy.
>
> To address this question, we compare the latency of InPars (Bonifacio et al., 2022), DocGen, and DocGen-RL in the following table. Please note that we report two variants of InPars: (i) the original InPars (Bonifacio et al., 2022) implementation which adopts GPT-J-6B, and (ii) a more efficient variant which adopts BLOOM-560M.
>
> | Data Augmentor                  | LLM | Latency     |  nDCG@10 on NQ
> |---------------------------------|----------------------|-------------|-------------|
> | InPars (Bonifacio et al., 2022) | GPT-J-6B             | 237 seconds | 0.335 |
> | DocGen                          | BLOOM-560M           | 107 seconds |  0.467 |
> | DocGen-RL                       | BLOOM-560M           | 107 seconds | 0.517 |
> | InPars (replicated by us)       | BLOOM-560M           | 72 seconds  | 0.337 |
>
>
> From the results, we see that both DocGen and DocGen-RL are over twice as efficient as InPars-GPT in inference (107 seconds vs. 237 seconds). However, they are less efficient than InPars-BLOOM (72 seconds vs. 107 seconds). We believe that the additional computational costs can be justified by the large performance gain that both models achieve (nDCG@10 on NQ: 0.517 (DocGen-RL) vs. 0.337 (InPars-BLOOM)).
>
> Regarding *"I wonder how much additional computational cost are brought by adopting RL policy."*: Compared to the DocGen, the additional RL training for DocGen-RL can be done offline and requires less than 18 hours of training with a single Nvidia P6000 GPU. This can be considered relatively low, especially when compared with the training cost of large language models. Moreover, it is noteworthy to mention that the inference times of DocGen and DocGen-RL are equal, since they both have the same architecture, while DocGen-RL is significantly more effective.

---

### Meta-Review · Area_Chair_2Ud1 · 2023-09-19

**Recommendation:** 4

**Metareview:**

This paper introduces DocGen, a new paradigm of generating synthetic documents from queries to augment the data for ranking models. An extended version DocGen-RL is further proposed to enhance the relevance between generated synthetic documents and queries using reinforcement learning. All reviewers find this paper strong and exciting with minor concerns on including more experimental analysis.

---

### Decision · Program_Chairs · 2023-10-07

**Decision:**

Accept-Main

**Comment:**

This paper introduces DocGen, a new paradigm of generating synthetic documents from queries to augment the data for ranking models. An extended version DocGen-RL is further proposed to enhance the relevance between generated synthetic documents and queries using reinforcement learning. All reviewers find this paper strong and exciting with minor concerns on including more experimental analysis.